# Expression and Prognostic Evaluation of the Receptor Tyrosine Kinase MET in Canine Malignant Melanoma

**DOI:** 10.3390/vetsci10040249

**Published:** 2023-03-26

**Authors:** Karen Koo, Arno Wuenschmann, Aaron Rendahl, Kyu Young Song, Colleen Forster, Amber Wolf-Ringwall, Antonella Borgatti, Alessio Giubellino

**Affiliations:** 1Department of Veterinary Clinical Sciences, College of Veterinary Medicine, University of Minnesota, St. Paul, MN 55455, USA; 2Veterinary Cancer Group, Torrance, CA 90504, USA; 3Veterinary Diagnostic Laboratory, Department of Veterinary Population Medicine, University of Minnesota, St. Paul, MN 55455, USA; 4Department of Veterinary and Biomedical Sciences, College of Veterinary Medicine, University of Minnesota, St. Paul, MN 55455, USA; 5Department of Laboratory Medicine and Pathology, University of Minnesota Medical School, Minneapolis, MN 55455, USA; 6Bionet, University of Minnesota, Minneapolis, MN 55455, USA; 7Masonic Cancer Center, University of Minnesota, Minneapolis, MN 55455, USA; 8Clinical Investigation Center, University of Minnesota, Minneapolis, MN 55455, USA

**Keywords:** MET, tyrosine kinase receptor, prognostic value, canine malignant melanoma, metastasis

## Abstract

**Simple Summary:**

Melanoma is a cancer of cells that produce melanin pigment, and it can develop in dogs as well as in humans. Canine melanoma has a high risk of metastasis and is resistant to systemic chemotherapy. Therefore, alternative effective systemic treatment is urgently needed. Abnormal expression and activation of the receptor tyrosine kinase MET is reported in many human cancers, including melanoma. However, the role of MET in canine melanoma is unknown. The purpose of this study is to confirm MET expression in canine melanomas and determine if the levels of expression correlate with prognosis. A review of 30 canine melanoma cases confirmed mild MET expression in more than 50% of these samples. While the levels of MET expression were not significantly associated with key histologic features or survival, a difference in time to metastasis to lymph nodes versus other organs was noted. This may suggest a role for MET expression in the pattern of metastasis, which may in turn help with patient stratification and treatment recommendation.

**Abstract:**

The overexpression and activation of the MET receptor tyrosine kinase has been identified in many human malignancies, but its role in canine cancer has only been minimally investigated. In this study we evaluated the expression of MET in two canine malignant melanoma (CMM) cell lines as well as in 30 CMM tissue samples that were collected from the clinical service at our institution. We were able to confirm the expression of the MET protein in both melanoma cell lines, and we demonstrated MET activation by its ligand, HGF, through phosphorylation, in Western blot analysis. We were also able to demonstrate, by immunohistochemistry, the expression of MET in 63% of the tumor tissue samples analyzed, with the majority demonstrating a relatively low expression profile. We then evaluated the association of MET expression scores with histologic parameters, metastasis, and survival. While statistically significant associations were not found across these parameters, an inverse relationship between MET expression levels and time to lymph node versus distant metastasis was suggested in our cohort. These findings may require assessment in a larger group of specimens to further evaluate the role of MET expression in the homing of metastasis in lymph nodes versus that in distant organs.

## 1. Introduction

Canine malignant melanoma (CMM) is an aggressive tumor with a high degree of local invasiveness and metastatic propensity. The reported one-year survival rate for dogs treated with surgery alone is about 20% [1]. Positive prognosticators include if the size of the tumor is smaller than 8 cm^3^ [2], an absence of metastasis, and rostral or lip locations, which in combination may provide a longer median survival of 1 to 2 years [3].

Human and canine melanoma, particularly mucosal and oral subtypes (oral, anorectal, vulvovaginal), share similar clinical and molecular similarities and are generally resistant to traditional chemotherapy [4,5,6,7]. Mucosal melanoma represents about 0.8–3.7% of all human melanomas, with an increased incidence in Asian and black populations [8]. About one third of cases have lymph node involvement at the time of diagnosis and the lungs are the most common site of distant metastasis [1]. Clinically, mucosal melanoma in humans carries a poor prognosis with a 5-year survival rate of only 25–33%, depending on disease stage and location [8].

In dogs, oral melanoma is the most common oral malignancy with higher reported incidences in breeds such as cocker spaniel, miniature poodle, Anatolian sheepdog, Gordon setter, chow chow, and golden retriever [9]. While surgery and coarsely fractionated radiation therapy can be effective at achieving local tumor control, these standardized treatments provide minimal stage-dependent clinical benefits, and death is usually due to metastatic disease. Current systemic therapy options for CMM, including chemotherapy or a commercially available xenogeneic DNA vaccine, have not been able to consistently show control of the progression of disease or extend survival [10,11]. This gap in therapeutic efficacy highlights the need to identify new targets in the treatment of metastatic CMM and to uncover biomarkers that can predict tumor response to novel treatment strategies.

The MET protooncogene is a surface receptor tyrosine kinase that is expressed on epithelial cells of many organs in physiologic conditions. After binding to the hepatocyte growth factor (HGF), its native and selective ligand, MET activates multiple downstream signaling pathways that result in cell proliferation, motility, and survival. Aberrant activation of the MET/HGF signaling pathway, commonly in both a ligand-dependent autocrine and paracrine mechanism or in a ligand-independent fashion, has been found in many human malignancies, such as papillary renal carcinoma, small cell lung cancer, and cutaneous melanoma [12,13,14,15]. In addition to its contribution to tumorigenesis, the level and pattern of MET expression are reportedly associated with high mitotic activity, tumor thickness, lymphatic and vascular invasion, nodal and/or visceral metastasis, and poor clinical outcomes in human cutaneous melanoma patients [12,16]. Moreover, MET activation is associated with resistance to targeted therapies with EGFR and VEGFR inhibitors [17] in other malignancies.

Given that the genetic landscape and mutations that have been identified in canine mucosal melanoma resemble that of human mucosal and acral melanoma, spontaneously occurring oral melanoma in dogs is a potential comparative oncology model for humans [1,8,18,19]. However, there have been no published studies to date investigating the correlation of MET expression with clinical outcomes in CMM. While MET inhibitors have shown some promise in controlling disease progression in human melanoma [20,21,22,23], similar targeted therapeutic options in dogs have not been investigated and remain an unmet need.

Here we investigate the expression of MET in melanoma cell lines and tissue samples of melanoma from our clinical service and correlate the expression scores with histologic parameters, metastatic status, and survival outcomes.

## 2. Materials and Methods

### 2.1. Case Selection and Clinical Data Collection

A search of the University of Minnesota Veterinary Medical Center (VMC) patient database from 2004 to 2019 for client-owned canine patients with documented diagnoses of melanoma yielded 517 cases, of which 175 cases had histopathology reports available for review. Cases were excluded for further analysis if the patient did not have a malignant melanoma as the final diagnosis, underwent any form of treatment before surgery, was considered to have a recurrent tumor based on the medical records, or if the surgery was performed at the referring veterinarian where tissue samples were not available for further evaluation. After reviewing the medical records and histopathology reports, 79 cases were identified as naïve canine malignant melanoma but only 37 cases had matching tissue samples available from Marshfield Laboratory, now a part of Idexx Laboratories. Recuts of tumor tissues were independently evaluated by a board-certified dermatopathologist (A.G.) and a veterinary anatomic pathologist (A.W.), and seven of those cases did not have evidence of tumor on recut so they were excluded from the final analysis. A total of 30 cases were included in the study which consisted of 20 oral malignant melanomas, 5 acral, and 5 cutaneous malignant melanomas.

Clinical data collected for each case included breed, gender, weight, age at diagnosis, date of diagnosis, tumor type (oral, acral, and cutaneous), treatment pursued, status and date of lymph node and/or distant metastasis, last contact date, and date and cause of death if available. Pertinent data that are presented in this study are available in Appendix A.

Informed consent was obtained from all of the University of Minnesota VMC clients for use of patients’ tissues for scientific study and teaching purposes. Only de-identified, archival, diagnostic material was used for this study so institutional review board or ethics committee approval were not applicable.

### 2.2. Western Blot Analysis

To confirm the expression of MET protein and MET phosphorylation after HGF stimulation, Western blots were performed using two CMM cell lines (TLM-1 and CMGD-2). The canine osteosarcoma cell line OSCA-8 (previously reported to express MET) was used as the positive control [24]. Sample preparation and Western blotting analysis were carried out according to standard laboratory protocols. Briefly, harvested cells were washed twice with PBS at 4 °C, and 0.1 mL of lysis buffer (50 mM Tris-HCl [pH 7.5], 150 mM NaCl, 0.25% sodium deoxycholate, 0.1% Nonidet P-40, 0.1% Triton X-100, 50 mM NaF, 1 mM dithiothreitol, 0.5 mM phenylmethylsulfonyl fluoride, 50 mM sodium pyrophosphate, 10 mM sodium vanadate, and 1X protease inhibitor cocktail [Roche]) was added directly to the culture dishes. After centrifugation, the supernatant was transferred to a new tube and SDS-PAGE sample buffer was added to the supernatant. Approximately 30 ug of protein from each lysate was resolved by SDS-PAGE using a 4–20% polyacrylamide gradient gel (Bio-Rad, Hercules, CA, USA). Gels were electroblotted onto polyvinylidene difluoride membranes (Amersham Bioscience, Amersham, UK) in transfer buffer (48 mM Tris-HCl, 39 mM glycine, and 20% methanol). Membranes were blocked in blocking solution (5% dry milk and 0.1% Tween 20 in Tris-buffered saline) overnight at 4 °C. Immunoblotting with MET (R&D systems, Cat. No. AF276), Phospho-MET (Cell Signaling Technology, Danvers, MA, USA; Tyr1234/1235, Cat. No.3077), Phospho-MET (Cell Signaling Technology, Tyr1349, Cat. No. 3121), and β-actin (Cell Signaling Technology, Cat. No. 3700) were performed according to the manufacturer’s instructions. Signals were detected using ECL substrate (ThermoFisher, Waltham, MA, USA), according to the manufacturer’s instructions. Western blot bands were analyzed with the Image StudioTM Lite software (LI-COR, Lincoln, NE, USA).

### 2.3. Validation of MET Antibody for Immunohistochemistry (IHC)

To prepare cell pellet blocks, cells were grown in T-75 flasks to near confluence, washed three times with PBS and fixed with 10% neutral-buffered formalin (NBF) in flasks overnight. After fixation, the cells were gently scraped and transferred to a conical tube and centrifuged to ~300× *g* for 5 min at room temperature (RT). Formalin supernatant was then removed, and the pellets were washed with PBS. Finally, the cell pellets were re-suspended in 80% ethanol and paraffin embedded in cell blocks. To validate the MET antibody (Met (D1C2) XP^®^ Rabbit mAb #8198, Cell Signaling Technology, Danvers, MA, USA) to be used in immunohistochemistry (IHC), cell pellets were generated from the three canine cell lines and processed by BioNet according to standard laboratory protocols.

### 2.4. IHC of Canine Melanoma Samples

Unstained tissue sections (4 µm) of the 30 identified cases were de-paraffinized and rehydrated using standard methods. For antigen retrieval, the slides were incubated in 6.0 pH buffer (Reveal Decloaking reagent, Biocare Medical, Concord, CA, USA) in a steamer for 30 min at 95–98 °C, followed by a 20-min cool down period. Subsequent steps were carried out manually. Endogenous peroxidase activity was quenched by slide immersion in 3% hydrogen peroxide solution (Peroxidazed, Biocare) for 10 min followed by TBST rinse. A serum-free blocking solution (Punisher, Biocare Medical, Concord, CA, USA) was placed on sections for 30 min. The blocking solution was removed and the slides were incubated in anti-MET (clone D1C2 XP(R)) (Cell Signaling, Danver, MA, USA; 1:50) diluted in 10% Sniper blocking solution/90% TBST overnight at 4 °C. On the following day, the slides were rinsed in TBST and biotinylated anti-rabbit (Vector Labs 1:200) was applied. The slides were incubated for 30 min at room temperature. Again, the slides were rinsed with TBST and 4+Streptavidin AP Label (Biocare Medical, Concord CA) was applied for 30 min. All the slides proceeded with TBST rinse and detection with Warp Red Chromogen (Biocare Medical, Concord, CA) using the manufacturer’s specification for 7 min at room temperature. The slides were then rinsed for 5 min in running tap water and counterstained with CAT Hematoxylin diluted 1:2 (Biocare Medical, Concord, CA) for 5 min, air-dried, dipped in xylene, and cover slipped.

### 2.5. Histologic Evaluation

Hematoxylin and eosin (H&E) and MET-stained canine melanoma samples were reviewed independently by a board-certified dermatopathologist (A.G.) and a veterinary anatomic pathologist (A.W.). The histologic data that were evaluated and collected included mitotic index (defined as mitotic count per 10 high-power fields), degree of pigmentation (<50% or >50%), presence of ulceration (Y or N), presence of necrosis (Y or N), presence of deep inflammation (Y or N), and Breslow thickness in millimeters. In human cutaneous melanocytic tumors, Breslow thickness is the measurement from the top of the granular layer of the epidermis or base of the ulceration to the deepest invasive cell across the base of the tumor. It is one of the most important independent prognostic indicators with thinner melanomas conferring a higher disease-free survival. While Breslow thickness has not been widely utilized in veterinary medicine, Silvestri et al. reported that tumor thickness was a prognostic indicator in canine melanocytic neoplasms, with cutoffs of 9.5 mm and 7.5 mm conferring a higher hazard for shorter overall survival and disease-free intervals, respectively [25].

The MET score was assigned to each sample based on the intensity of cell membrane staining and was independently evaluated by each of the two reviewers. The scores ranged from 0 to 2, with 0 for negative immunopositivity, 1 for mild/moderate, and 2 for marked immunopositivity. All of the slides were processed together to eliminate variations in staining between batches. A consensus was reached when there were discrepancies in the assigned scores. Normal canine liver tissue was used as a positive control. Representative histologic pictures after H&E and MET staining are shown in Figure 1.

### 2.6. Statistical Analysis

Several analyses were performed to assess the MET prognostic value. For binary values (pigmentation, ulceration, necrosis, inflammation), the Cochran–Armitage trend test was used; for numerical values (Mitotic index), Spearman’s rank correlation was used; and for censored data (Breslow) and time to event data, the log-rank test for trend was used. Analyses were also performed to assess differences between tumor types. For binary values, Fisher’s exact test was used; for numerical values, the Kruskal–Wallis test was used; and for censored data and time to event data the log-rank test was used. Time to lymph node metastasis and distant metastasis were considered censored at either time to last follow-up or time of death; therefore, these should be interpreted as the time to these events if they had not died first. Time to death was censored if they were lost to follow-up while still alive. For binary variables, counts and percentages are reported; for continuous data, medians and ranges; for censored data (Breslow), the estimated percent >5 from the Kaplan–Meier curve and the number of uncensored data; and for time to event data, Kaplan–Meier curves are shown. All calculations were performed in R version 4.0.2 (22 June 2020).

## 3. Results 

### 3.1. Sample Population Characteristics

A total of 30 cases of CMM were selected from our database, including 18 female dogs (60.0%) with an average weight of 30.1 kg (range 3.2–68.8 kg) and an average age of 10 years at the time of diagnosis, and 12 male dogs (40.0%) with an average weight of 35.5 kg (range 12.3–47.2 kg) and an average age of 9.5 years at time of diagnosis. The three most common breeds were Labrador retriever (*n* = 6, 20.0%), German shepherd (*n* = 4, 13.3%), and golden retriever (*n* = 3, 10.0%). A total of 15 dogs (50.0%) had reported lymph node and/or distant metastasis, of which nine dogs (30.0%) had distant metastasis only, three dogs (10.0%) had both confirmed or suspected lymph node and distant metastasis, and two dogs (6.7%) had lymph node involvement only. Additional clinical characteristics of these cases can be found in Appendix A.

A total of 27 dogs (90.0%) received some form of treatment after diagnosis. Of these, 11 (36.7%) and 8 (26.7%) of these dogs underwent surgery plus adjuvant melanoma vaccine or radiation therapy, respectively. The breakdown of the treatment options elected can be found in Appendix A.

### 3.2. MET Protein Expression and Activation in Canine Melanoma Cell Lines

To confirm the expression of MET protein and MET phosphorylation after HGF stimulation, Western blots were performed using two melanomas and one osteosarcoma cell lines. The results of the Western blots confirmed MET protein expression in all three cell lines. Critical tyrosine residues in the kinase domain and a multifunctional docking site of the MET receptors, namely tyrosine 1234/1235 and 1349 which modulate enzyme activity and recruit transducers and adaptors after MET activation, respectively, were also found to be phosphorylated appropriately after stimulation with recombinant human HGF (Appendix A). These findings confirm MET tyrosine kinase activation in all three canine cell lines.

To validate the use of the same antibody for immunohistochemistry, we created formalin-fixed and paraffin-embedded cell blocks from pellets of the same three cell lines. Immunohistochemistry with the monoclonal MET-specific antibody was positive in the three cell lines, effectively validating this antibody for immunoprofiling of tissue sections from our cohort of canine melanoma samples (Appendix A).

### 3.3. Histopathologic Characteristics and Survival Data of the Sample Population

The parameters that were evaluated during histopathologic review of the 30 CMM cases included the degree of pigmentation, presence of ulceration, presence of necrosis, presence of deep inflammation, mitotic index (MI), and Breslow thickness. A summary of the results by tumor type is available in Table 1. The median MIs for oral, acral, and cutaneous melanoma were 2.5 (range 0–10), 3 (range 1–4), and 3 (range 0–4), respectively. The median Breslow thickness for oral, acral, and cutaneous melanoma were 6.4 mm (range 2–13 mm), 6 mm (range 3.5–10 mm), and 5 mm (range 2–20 mm), respectively. No statistically significant differences between the different subtypes of melanoma were found.

Survival outcome data were available for 17 dogs (14 oral, 2 acral, 1 cutaneous, 56.7%) while 11 of the dogs were lost to follow-up (36.7%). There were two dogs that were alive at the end of data collection period (6.6%). The median survival time for all dogs with oral melanoma was 457 days (range 41–1023 days). The two dogs with acral melanoma survived 829 day and 290 days, respectively. There was one dog with cutaneous melanoma that had a survival time of 329 days.

### 3.4. MET Protein Expression in Canine Melanoma Tissue Samples

We then performed immunohistochemical assessment of MET expression on the CMM clinical tissue samples. Of all 30 cases of CMM that were reviewed, 16 (53.3%) had an MET score of 1, and 3 of the cases (10.0%) had an MET score of 2. The rest of the 11 cases (36.7%) were negative for MET expression. When the MET scoring was broken down by tumor types, an MET score of 1 was the most common among all the subtypes, including 10 oral melanomas (50.0% of all oral cases) and 3 of each acral and cutaneous melanomas (60.0% of each subtype of melanoma) (Table 2).

We then evaluated the relationship between histopathologic features and MET expression for all canine melanomas. Using univariate analysis, we did not find statistical significance for any of the parameters evaluated (Appendix A); similar results were obtained when we analyzed the largest subgroup of oral canine melanomas specifically (Appendix A).

Finally, we investigated the relationship between MET expression and the time to metastasis or death (i.e., survival time). The association of MET score with time to disease metastasis, including lymph node metastasis or distant spread, and death for all types of melanomas was not found to be statistically significant (Figure 2a–d). Similar results were noted when evaluating oral melanomas alone (Figure 2e–g).

While no statistically significant associations were identified across these parameters, a possible inverse relationship between the MET score and time to lymph node metastasis was noted (Figure 2a,e). On the other hand, this inverse MET score relationship was not as evident for the time to distant metastasis (Figure 2b,f). This possible association and difference in lymph node verses distant metastasis will require assessing a larger group of specimens to confirm potential clinical implications.

## 4. Discussion

In this study, we aimed to investigate MET protein expression and activation in canine melanoma cell lines as well as investigate MET expression in tissue samples of canine melanoma from our clinical practice in a tertiary care academic veterinary practice. Canine melanoma, unlike human melanoma, is most commonly a mucosal or acral disease, rather than cutaneous. Interestingly, there are several similarities between oral and acral melanoma in humans and the corresponding disease in dogs, making the latter an attractive comparative model [1,8,18,19].

Our current knowledge of the molecular underpinning of melanoma in dogs is piecemeal at best; this lack of knowledge is reflected by the limited therapeutic options in canine patients, accompanied by a lack of robust markers for disease progression and prognosis. Thus, development of novel predictive and prognostic biomarkers is a priority.

Receptor tyrosine kinases (RTKs) are attractive targets for oncologic therapy. Several selective inhibitors have been developed over the past two decades. Among these RTKs, the MET receptor appears to play a role in the pathogenesis and progression of melanoma in humans [12,16,26]. Thus, we investigated the expression of MET protein in canine melanoma and its association with histologic characteristics and clinical outcomes.

In this study, we confirmed MET protein expression in canine melanoma cell lines as well as tumor tissues. We also demonstrated the activation through phosphorylation of MET in key tyrosine kinase domains in canine melanoma cell lines through Western blots. This includes the phosphorylation of tyrosine residues 1234/1235 in the kinase domain of MET with consequent activation of the catalytic activity of the receptor and activation through phosphorylation of tyrosine residue 1349 in the multifunctional docking site in the c-terminus of the receptor. This domain is particularly important for connecting the activated MET receptor to downstream signaling through the use of adaptor proteins and enzymatic reactions, which will ultimately lead to relevant gene transcriptions in the nucleus. Thus, we have demonstrated that the MET receptor is present and can be activated in canine melanoma cell lines.

The expression of MET in canine melanoma tissue was confirmed through immunohistochemistry with a validated antibody. We confirmed MET immunopositivity in canine malignant melanoma tissues in 63% of the specimens. This demonstrates that MET is present in primary melanoma tumors in dogs, although the expression appears to occur at low levels, as demonstrated by the presence of +1/mild immunopositivity in the majority of the specimens.

MET expression has been reported in the literature in other canine tumors, although reports are sporadic. In 2005, Liao et al. evaluated the expression and function of MET in 25 canine cancer cell lines including osteosarcoma and melanoma [24]. All cell lines of the study expressed mRNA for MET. Moreover, MET protein expression and MET phosphorylation after HGF stimulation were also documented in canine osteosarcoma and melanoma cell lines in that study, confirming activation of the receptor in vitro.

Overall, we did not observe statistical significance in the associations of MET expression with histopathologic parameters that are commonly investigated and recorded during histopathologic evaluation. However, we made some interesting observations that are important to outline here. By comparing the time to event curves of all canine melanoma patients with the three levels of MET expression, we noted that the line representing negative MET staining (“CMET = 0”) is consistently lower than the one for mild immunopositivity (“CMET = 1”) for time to lymph node metastasis (Figure 2a). This suggested an overall shorter time to lymph node metastasis for those cases with negative MET expression compared to that with mild MET expression. On the contrary, the median time to distant metastasis for those with negative MET expression was longer than that with mild expression (Figure 2b), which is in line with the known role of MET in the development of metastatic disease at distant sites, as demonstrated in several malignancies [26,27,28,29,30]. This difference in MET expression levels in time to lymph node metastasis versus that in distant metastasis was also noted when evaluating canine oral melanomas specifically (Figure 2e,f). This finding is interesting because it proposes the hypothesis that MET expression has different roles in the homing of metastasis in lymph nodes versus that in distant organs. If confirmed in a larger sample, this observation may be clinically relevant in identifying groups that are at risk of lymph node versus distant metastasis.

Based on a comprehensive review of prognostic factors in canine melanoma, the mitotic index has been previously identified as an independent prognostic factor [31]. An MI >= 4 for canine oral and lip melanocytic neoplasm was associated with increased risk of death within one year of diagnosis, with a 90% sensitivity and 84% specificity. For canine cutaneous melanoma, an MI >= 3 was statistically correlated with a low 2-year survival rate. For the current study, the median mitotic indices for oral, acral, and cutaneous melanomas were 2.5, 3, and 3, respectively. These values are lower than the established cut-off for aggressive phenotypes. Thus, our cases may have been overall less aggressive, which may account for the lower level of MET detected in our samples. Given the hypothesis that MET expression confers a more metastatic phenotype and hence more aggressive profile, including cases that are less aggressive, may have undermined otherwise significant relationships between MET expression and prognosis in this study.

Similar to other retrospective studies, a small sample size and incomplete data from loss to follow-up are among the limitations of the current study. Of the 30 cases with histologic diagnosis of melanoma and tissue samples available, 13 were censored with 11 lost to follow-up and 2 dogs alive at the end of data collection. The inconsistency in the follow-up of these cases also affected the reported time to metastasis and death—biases that a larger scale prospective study with set recheck schedules and re-staging requirements can overcome. Despite the limitations in the data collected, we were able to retrieve clinical and histopathologic information as reported in Table 1 and Appendix A. This has allowed us to investigate correlations of MET expression with several parameters, even though we were not able to find a definitive statistical significance for any of the parameters that were evaluated. The other limitation of this study is the relatively limited number of samples for each of the categories of melanoma evaluated. We originally were able to find more cases in our initial search, but we elected to include only the cases for which we had the most complete and reliable data.

In conclusion, this is the first study to evaluate the level of MET expression with prognosis and survival in canine melanoma tissue samples. These findings may require assessment in a larger group of specimens to confirm the clinical implications of MET expression assessment in CMM.

## Figures and Tables

**Figure 1 vetsci-10-00249-f001:**
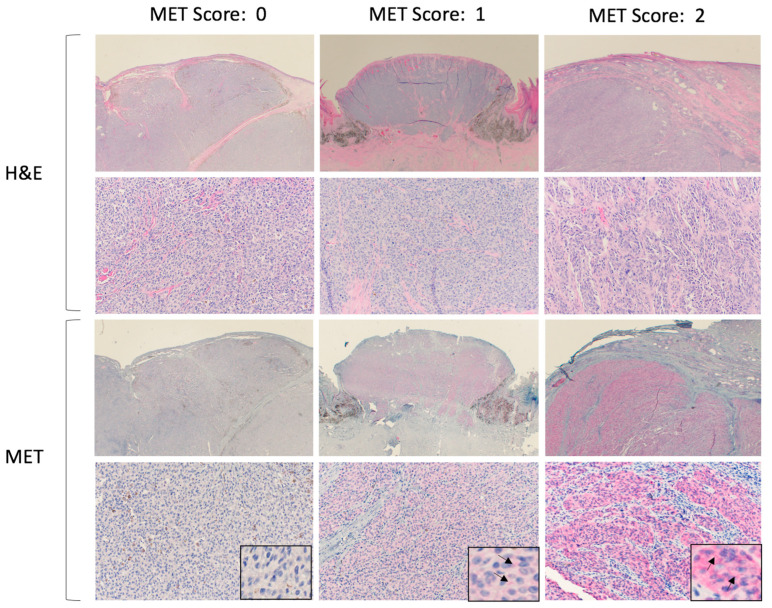
Representative histologic samples of canine malignant melanoma tissue after H&E and c-MET immunostaining. A score of 0 is for negative immunopositivity, 1 for mild/moderate, and 2 for marked immunopositivity. Insets at high magnification for MET immunostaining highlight membranous staining (arrows).

**Figure 2 vetsci-10-00249-f002:**
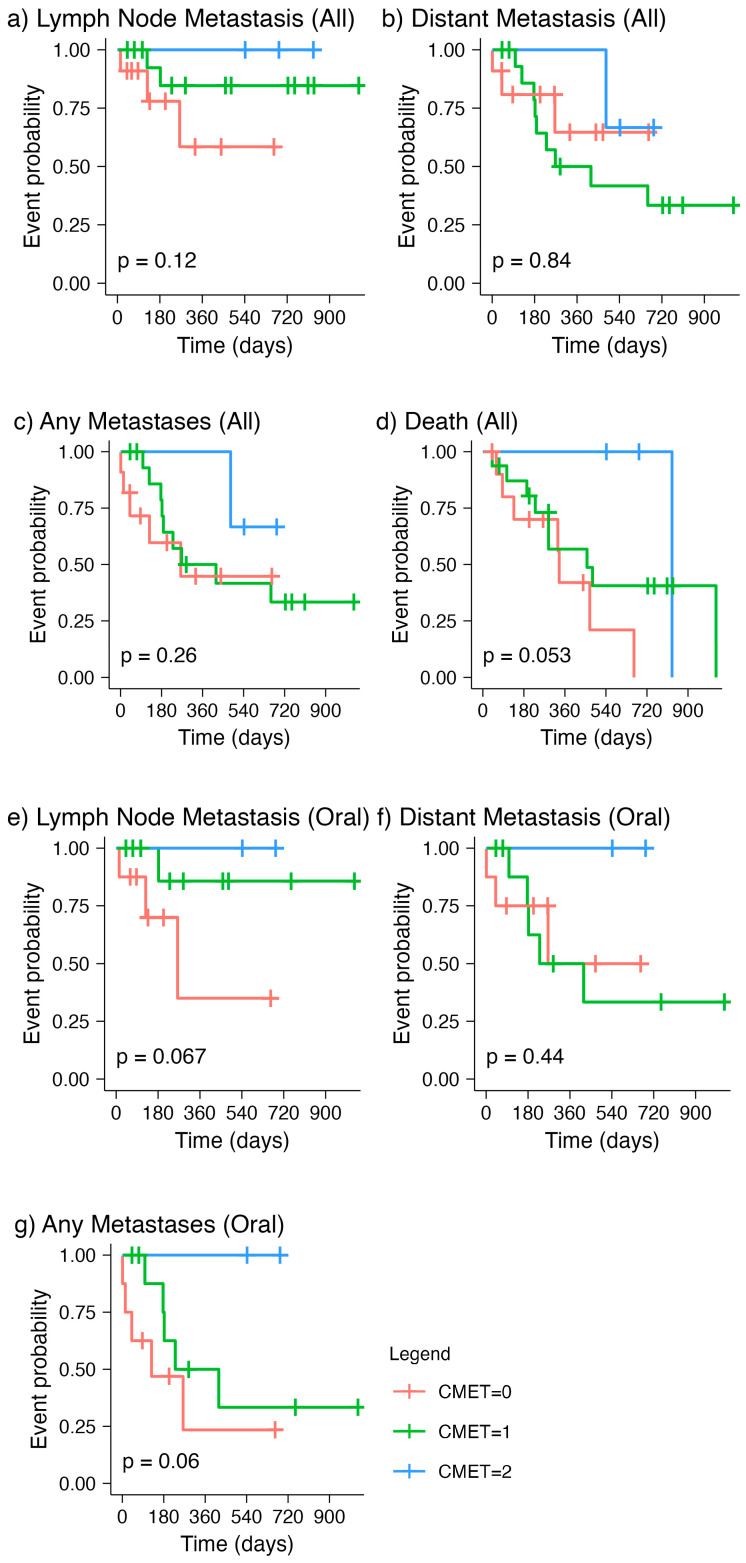
Kaplan–Meier curves by Met scores for time to (**a**) lymph node metastasis, (**b**) distant metastasis, (**c**) any metastases, (**d**) death for all canine melanomas, and for time to (**e**) lymph node metastasis, (**f**) distant metastasis, and (**g**) any metastases for canine oral melanomas.

**Table 1 vetsci-10-00249-t001:** Summary of histologic features evaluated by type of canine melanoma.

Histologic Feature	Values	Tumor Type			*p*-Value
		Oral	Acral	Cutaneous	
Degree of pigmentation	<50%	12	5	2	0.18
	>50%	8	0	3	
Presence of ulceration	N	4	2	2	0.57
	Y	16	3	3	
Presence of necrosis	N	16	3	4	0.81
	Y	4	2	1	
Presence of deep inflammation ^1^	N	8	2	3	1.00
	Y	8	2	2	
Median mitotic index ^2^	--	2.5	3	3	0.73
Median minimum Breslow thickness (mm)	--	6.4	6	5	0.72

^1^ 5 cases (4 orals and 1 acral) were not able to be classified due to lack of inflammation or tumor extension to tissue margin. ^2^ Mitotic index = mitotic count per 10 high power fields

**Table 2 vetsci-10-00249-t002:** Count of Met score by melanoma tumor type.

Met Score	Tumor Type			Total
	Oral	Acral	Cutaneous	
0	8	1	2	11
1	10	3	3	16
2	2	1	0	3

## Data Availability

The data presented in this study are available in the Appendix A.

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
