# Peer review of "Expression and Prognostic Evaluation of the Receptor Tyrosine Kinase MET in Canine Malignant Melanoma"

_vetsci, 2023, doi:10.3390/vetsci10040249_

Round 1
Reviewer 1 Report
The prognostic evaluation of the receptor tyrosine 2 kinase MET in canine malignant melanoma was investigated by the researchers in this study.
The necessary literature on the subject was searched and the applied methods were given in detail. The findings obtained in the study are presented in detail with graphics, tables and histopathological pictures. The findings obtained at the end of the study were compared with other studies on the subject.
The researchers investigated that expression of MET in melanoma cell lines and tissue samples of melanoma from their clinical service and correlate the expression scores with histologic parameters, metastatic status and survival outcomes. Also, the researchers did not observe statistical significance in the associations of MET expression with histopathologic parameters that are commonly investigated and recorded during histopathologic evaluation. On the other hand, this is the first study to evaluate the level of MET expression with prognosis and survival in canine melanoma tissue samples.
Author Response
Thank you for your time in reviewing the manuscript and your comments.
Reviewer 2 Report
This is a very interesting paper that deserves to be published after minor revisions.
1 - Please clarify photos to show specifically the membrane immunostaining of the used antibodies.
2 - Also please enlarge the KM graphics... bigger legends improve improve the readability of the paper
3- The authors maybe can consider the recent review to include in the references: Sabbah M, Najem A, Krayem M, Awada A, Journe F, Ghanem GE. RTK Inhibitors in Melanoma: From Bench to Bedside. Cancers (Basel). 2021 Apr 2;13(7):1685.
Author Response
- We have included insert in Figure 1 to illustrate membrane immunostaining (highlighted by arrows) with the antibody used in the study.
- KM graphs have been modified to include larger legends to improve readability.
- We have read with interest the suggested recent article suggested by one of the reviewers. Thank you for your suggestion. This article was helpful in supporting our discussion that RTKs, including MET receptor, are becoming therapeutic targets in oncology. It is therefore included in our discussion and cited as a reference.